# Mathematical Justification of Hard Negative Mining via Isometric Approximation Theorem

**Albert Xu, Jhih-Yi Hsieh, Bhaskar Vundurthy, Nithya Kemp, Eliana Cohen, Lu Li, & Howie Choset**
Robotics Institute
Carnegie Mellon University
Pittsburgh, PA 15232, USA
{albertx, jhihyih, pvundurt}@andrew.cmu.edu

## Abstract

In deep metric learning, the triplet loss has emerged as a popular method to learn many computer vision and natural language processing tasks such as facial recognition, object detection, and visual-semantic embeddings. One issue that plagues the triplet loss is network collapse, an undesirable phenomenon where the network projects the embeddings of all data onto a single point. Researchers predominately solve this problem by using triplet mining strategies. While hard negative mining is the most effective of these strategies, existing formulations lack strong theoretical justification for their empirical success. In this paper, we utilize the mathematical theory of isometric approximation to show an equivalence between the triplet loss sampled by hard negative mining and an optimization problem that minimizes a Hausdorff-like distance between the neural network and its ideal counterpart function. This provides the theoretical justifications for hard negative mining's empirical efficacy. Experiments performed on the Market-1501 and Stanford Online Products datasets with various network architectures corroborate our theoretical findings, indicating that network collapse tends to happen when the batch size is too large or embedding dimension is too small. In addition, our novel application of the isometric approximation theorem provides the groundwork for future forms of hard negative mining that avoid network collapse.

## 1 Introduction

Research in deep metric learning investigates techniques for training deep neural networks to learn similarities and dissimilarities between data samples. This is typically achieved by learning a distance metric via feature embeddings in $\mathbb{R}^n$. Deep metric learning is commonly applied to face recognition Schroff et al. (2015); Liu et al. (2017); Hermans et al. (2017) and other computer vision tasks Tack et al. (2020); Chen et al. (2020a) where there is an abundance of label values.

Contrastive loss Hadsell et al. (2006) and triplet loss Schroff et al. (2015) are two prominent examples of deep metric learning, each with variants to address specific applications. For instance, SimCLR Chen et al. (2020a;b) is a recent contrastive loss variant designed to perform unsupervised deep metric learning with state-of-the-art performance on ImageNet Russakovsky et al. (2015). Ladder Loss Zhou et al. (2019), a generalized variant of the triplet loss, handles coherent visual-semantic embedding and has important applications in multiple visual and language understanding tasks Karpathy et al. (2014); Ma et al. (2015); Vinyals et al. (2014). Cross-level concept distillation Zheng et al. (2022) achieves state of the art performance on hierarchical image classification and dynamic metric learning. Given the success of metric learning in a wide range of applications, we see value in investigating its underlying theories. In particular, we focus on the triplet loss and present a theoretical framework which explains observed but previously unexplained behaviors of the triplet loss.

A triplet selection strategy is fundamental to any triplet loss-based deep metric learning Wu et al. (2017). This paper deals with hard negative mining, a triplet selection strategy that outperforms other mining strategies in a number of applications, for instance, person re-identification Hermans et al. (2017). In some scenarios, hard negative mining is known to suffer from network collapse,

a phenomenon where the network projects all data points onto a single point. Schroff et al. (2015) observe this effect in their experiments with a person re-identification dataset. On the other hand, Hermans et al. (2017) show that hard negative mining does not suffer from collapsed solutions for a similar dataset. These seemingly contradictory results showcase the need for a theoretical framework to explain the nature of hard negative mining and the root cause for any collapsed solutions.

There has been some prior literature investigating the phenomenon of network collapse. Xuan et al. (2020) show that hard negative mining leads to collapsed solutions by analyzing the gradients of a simplified neural network model. Levi et al. (2021) prove that, under a label randomization assumption, the globally optimal solution to the triplet loss necessarily exhibits network collapse. However, neither analysis offers sufficient explanation for why hard negative mining can work in practice Hermans et al. (2017); Faghri et al. (2017) and how one may reproduce such desirable behavior.

In this work, we explain why network collapse happens by using the theory of isometric approximation Vaisala (2002a) to draw an equivalence between the triplet loss with hard negative mining and a Hausdorff-like distance metric (Sec 3.2.1). On the Hausdorff-like metric, we observe that collapsed solutions are more likely when the batch size is large or when the embedding dimension is small. Our experiments with the person re-identification dataset (Market-1501 Zheng et al. (2015)) reconcile the findings of Schroff et al. (2015), where a batch size in the order of thousands led to network collapse, and Hermans et al. (2017), where a batch size of $N = 72$ showed no network collapse. We further support our predictions via experiments spanning three additional datasets (SOP Oh Song et al. (2016), CARS Krause et al. (2013), and CUB200 Wah et al. (2011)) and three different network architectures (ResNet-18, ResNet-50 He et al. (2016), GoogLeNet Szegedy et al. (2015), and a 2-layer convnet).

The paper is organized as follows. We begin with the definition of triplet loss with hard negative mining and then present the isometric approximation theorems in Section 2. In Section 3, we define the Hausdorff-like distance and outline a proof of its equivalence to the triplet loss with hard negative mining. This leads to a measure for network collapse followed by an illustration on how the network collapse is related to the batch size and embedding dimension. Section 4 demonstrates the validity of our theory with experiments on four datasets spanning three network architectures. Section 5 concludes the paper with possible future applications for our theory.

## 2 BACKGROUND AND DEFINITIONS

Let $\mathcal{X}$ be the data manifold and let $\mathcal{Y}$ be the classes with $|\mathcal{Y}| = c$ being the number of classes. Let $h : \mathcal{X} \rightarrow \mathcal{Y}$ be the true hypothesis function, or true labels of the data. Then the dataset consists of pairs $\{(x_k, y_k)\}_{k=1}^N$ with $x_k \in \mathcal{X}, y_k \in \mathcal{Y}$ and $y_k = h(x_k)$. We define the learned neural network as a function $f_\theta : \mathcal{X} \rightarrow \mathbb{R}^n$ which maps similar points in the data manifold $\mathcal{X}$ to similar points in $\mathbb{R}^n$.

As our paper focuses on metric learning, we define the similarity between embeddings to be the Euclidean distance

$$d_\theta(x_1, x_2) = ||f_\theta(x_1) - f_\theta(x_2)|| \tag{1}$$

where $x_1, x_2 \in \mathcal{X}$.

### 2.1 TRIPLET LOSS AND HARD NEGATIVE MINING

In this section, we discuss the triplet loss that considers triplets of data composed of the anchor ($x \in \mathcal{X}$), positive ($x^+$), and negative ($x^-$) samples, described in (2a) and (2b). The similarity relation (2a) requires that the anchor and positive samples must be of the same class, while the dissimilarity relation (2b) requires the anchor and negative must be of different classes.

$$x^+ \in \{x' \in \mathcal{X} | h(x) = h(x')\} \tag{2a}$$

$$x^- \in \{x' \in \mathcal{X} | h(x) \neq h(x')\} \tag{2b}$$

Restating the objective of supervised metric learning, the embedding of the anchor sample must be closer to the positive than the negative for every triplet. An example of a satisfactory triplet is shown

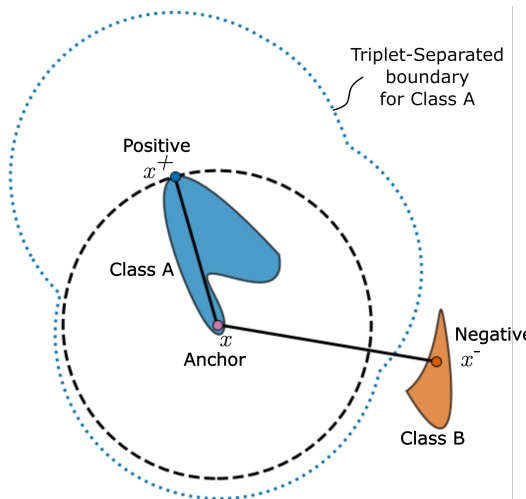

Figure 1: An example anchor, positive, and negative triplet. The blue dotted contour is the Triplet-Separated boundary for Class A. It is computed by considering inequality (5) for all points in Class A. Because Class B is outside the Triplet-Separated boundary for Class A, the triplet loss for this example is zero.

in Figure 1. Formally, we express this relation via (3), where $\alpha$ is the margin term.

$$d_\theta(x, x^+) + \alpha \le d_\theta(x, x^-) \quad \forall\, x, x^+, x^- \in \mathcal{X} \tag{3}$$

This leads to the definition of the triplet loss in (4).

$$\mathcal{L}_{\text{Triplet}} = \left[ d_\theta(x, x^+) - d_\theta(x, x^-) + \alpha \right]_+ \tag{4}$$

The function $[\,\cdot\,]_+ = \max(\cdot, 0)$ zeroes negative values in order to ignore all the triplets that already satisfy the desired relation.

**Definition 2.1.** $\alpha$-***Triplet-Separated***. *We refer to $m$ non-empty subsets $X^1, \cdots, X^m \subset \mathbb{R}^n$ as $\alpha$-**Triplet-Separated** if for every $X^i$ and $X^j$ with $i \ne j$ we have*

$$||x - y|| + \alpha \le ||x - z|| \quad \forall x, y \in X^i, \forall z \in X^j \tag{5}$$

*This property can be extended to a function $f_\theta : \mathcal{X} \to \mathbb{R}^n$ by checking whether the embedding subsets $X^i_{f_\theta}$ are $\alpha$-Triplet-Separated.*

$$X^i_{f_\theta} = \{ f_\theta(x) | x \in \mathcal{X}, h(x) = i \} \tag{6}$$

It is worth noting that $\mathcal{L}_{\text{Triplet}}(f_\theta) = 0$ if and only if $f_\theta$ is $\alpha$-Triplet-Separated. An example of two Triplet-Separated sets is shown in Figure 1.

As mentioned in Section 1, the triplet loss relies heavily on its triplet mining strategy to achieve its performance for two popularly accepted reasons: First, enumerating all $O(N^3)$ triplets of data every iteration would be too computationally intensive to be tractable. Second, improper sampling of triplets risks network collapse Xuan et al. (2020). Our work substantiates the use of hard negative mining, a successful triplet mining strategy, by characterizing conditions that lead to network collapse.

## 2.2 ISOMETRIC APPROXIMATION

We will present a novel application of the isometric approximation theorem Vaisala (2002a) in Euclidean subsets in order to mathematically justify hard negative mining. The isometric approximation theorem primarily defines the behavior of near-isometries, or functions that are close to isometries, as given by Definition 2.2.

**Definition 2.2.** $\varepsilon$-*nearisometry*. *Let $X$ and $Y$ be real normed spaces. A function $f : A \to Y$ where $A \subset X$ is called an $\varepsilon$-**nearisometry** ($\varepsilon > 0$) if*

$$\left| ||f(x) - f(y)|| - ||x - y|| \right| \leq \varepsilon, \ \forall \, x, y \in A \tag{7}$$

In other words, an $\varepsilon$-nearisometry is a function that preserves the distance metric within $\varepsilon$. The isometric approximation theorem seeks to determine how close $f$ is to an isometry, say $U : X \to Y$, as given by (8). Note $q_A(\varepsilon)$ is a function of $\varepsilon$ that is fixed for a given $A$ and is thus independent of $f$. Consequently, inequality (8) holds for all $\varepsilon > 0$ and all $\varepsilon$-nearisometries $f$.

$$||f(x) - U(x)|| \leq q_A(\varepsilon) \ \forall \, x \in A \tag{8}$$

Now consider the case where $X$ and $Y$ are n-dimensional Euclidean metric spaces, making $A \subset \mathbb{R}^n$. Then the following theorems and definitions Vaisala (2002a;b); Alestalo et al. (2001) prove that $q_A(\varepsilon)$ is linear in $\varepsilon$ given a thickness condition on the set $A$.

**Definition 2.3.** *Thickness*. *For each unit vector $e \in S^{n-1}$, define the projection $\pi_e : \mathbb{R}^n \to \mathbb{R}$ by the dot product $\pi_e(x) = x \cdot e$. The thickness of a bounded set $A$ is the number*

$$\theta(A) = \inf_{e \in S^{n-1}} \text{diam}(\pi_e A) \tag{9a}$$

$$\text{where } \text{diam}(X) = \sup_{r_1, r_2 \in X} ||r_1 - r_2|| \tag{9b}$$

**Theorem 2.4** (From **Theorem 3.3** Alestalo et al. (2001)). *Suppose that $0 < q \leq 1$ and $A \subset \mathbb{R}^n$ is a compact set with $\theta(A) \geq q \ diam(A)$. Let $f : A \to \mathbb{R}^n$ be an $\varepsilon$-nearisometry. Then there is an isometry $U : \mathbb{R}^n \to \mathbb{R}^n$ such that*

$$||f(x) - U(x)|| \leq c_n \varepsilon / q \ \ \forall x \in A \tag{10}$$

*with $c_n$ depending only on dimension.*

As this property depends entirely on the set $A$, we call Theorem 2.4 the $c$-**Isometric Approximation Property** ($c$-IAP) on set $A$ with $c = c_n \, \text{diam}(A)/\theta(A)$.

## 3 THEORETICAL CONTRIBUTIONS

### 3.1 OVERVIEW AND PROBLEM SETUP

From the background and definitions, the goal of the triplet loss is to learn a function $f_\theta$ such that the induced distance metric $d_\theta$ satisfies the property in (3). In this paper, we aim to justify the use of hard negative mining with the triplet loss for this task and offer theoretical explanations for why it sometimes leads to collapsed solutions. Furthermore, we wish to be able to predict when network collapse happens.

In this section, we first prove an equivalence between a Hausdorff-like distance (13) and the triplet loss with hard negative mining. Then, with the diameter of the network's embedding set as an indicator for network collapse ($\text{diam}(X_{f_\theta}) = 0$), we use the previous equivalence to show that network collapse happens when the batch size is large or when the embedding dimension is small.

### 3.2 EQUIVALENCE BETWEEN $d_{\text{HAUS}}$ AND TRIPLET LOSS

We begin with the definition of the Hausdorff-like distance $d_{\text{haus}}$, draw an equivalence to the isometric error $d_{\text{iso}}$, which is equivalent to the triplet loss within a constant factor. Then, we illustrate all three distance functions and their equivalence with a toy example.

#### 3.2.1 HAUSDORFF-LIKE DISTANCE $d_{\text{HAUS}}$

Reiterating the training objective from the problem setup, we aim to learn a function $f_\theta$ that is $\alpha$-Triplet-Separated (Definition 2.1). We restate this problem as a distance minimization problem, and prove that it is equivalent to hard negative mining with the triplet loss.

First we construct set of all functions $f : \mathcal{X} \to \mathbb{R}^n$ that are $\alpha$-Triplet-Separated and denote it with $\mathcal{F}_{TS}^\alpha$. We next construct the Hausdorff-like distance metric (denoted by $d_{\text{haus}}$) between these functions that compares the embedding subsets via the Hausdorff distance metric $d_H$.

$$X_f^i = \{f(x) | x \in \mathcal{X}, h(x) = i\} \tag{11}$$

$$d_{\text{haus}}(f_1, f_2) = \max_{i \in \mathcal{Y}} d_H(X_{f_1}^i, X_{f_2}^i) \tag{12}$$

One way to solve metric learning is to find the closest $f_\theta$ to any function in $\mathcal{F}_{TS}^\alpha$ as indicated by (13).

$$d_{\text{haus}}(f_\theta, \mathcal{F}_{TS}^\alpha) = \inf_{f_{\text{haus}}^* \in \mathcal{F}_{TS}^\alpha} d_{\text{haus}}(f_\theta, f_{\text{haus}}^*) \tag{13}$$

In this paper, we claim that the triplet loss with hard negative mining is equivalent to minimizing $d_{\text{haus}}(f_\theta, \mathcal{F}_{TS})$ within a constant factor (see Corollary 3.4).

### 3.2.2 ISOMETRIC APPROXIMATION APPLIED TO $d_{\text{HAUS}}$

In this section, we present Theorem 3.2 to show that minimizing the Hausdorff-like distance is equivalent to minimizing a difference in distance metrics, referred to as the **isometric error** (Definition 3.1).

**Definition 3.1.** *isometric error. For two functions $f, g : \mathcal{X} \to \mathbb{R}^n$, we define the isometric error $d_{iso}$ to be the maximum difference between their distance metrics.*

$$d_{iso}(f, g) = \sup_{x_1, x_2 \in \mathcal{X}} \left| ||f(x_1) - f(x_2)|| - ||g(x_1) - g(x_2)|| \right| \tag{14}$$

Similar to (13), we extend the definition of isometric error to $d_{iso}(f_\theta, \mathcal{F}_{TS}^\alpha)$ as follows:

$$d_{iso}(f_\theta, \mathcal{F}_{TS}^\alpha) = \inf_{f_{iso}^* \in \mathcal{F}_{TS}^\alpha} d_{iso}(f_\theta, f_{iso}^*) \tag{15}$$

**Theorem 3.2.** *$d_{haus}(f_\theta, \mathcal{F}_{TS}^\alpha)$ and $d_{iso}(f_\theta, \mathcal{F}_{TS}^\alpha)$ upper bound each other within a linear factor for all $f_\theta$ with some minimum thickness $\theta_*$.*

We present the proof for Theorem 3.2 in Appendix A. Theorem 3.2 shows that $d_{\text{haus}}$ and $d_{\text{iso}}$ are exchangeable as minimization objectives because they upper bound each other within linear factors. And as $d_{\text{iso}}$ is a difference of two distance functions, we can derive the triplet loss.

### 3.2.3 RECOVERING THE TRIPLET LOSS

In this section, we present Theorem 3.3 to show that the isometric error (Definition 3.1) is equivalent to the triplet loss sampled by hard negative mining.

**Theorem 3.3.** *The triplet loss sampled by hard negative mining and the isometric error $d_{iso}$ upper bound each other within a linear factor.*

We present the proof for Theorem 3.3 in Appendix B, proving that $d_{\text{iso}}$ is exchangable with the triplet loss sampled by hard negative mining. Thus from Theorems 3.2 and 3.3, we have Corollary 3.4.

**Corollary 3.4.** *The optimal solution to the triplet loss sampled by hard negative mining is equivalent to the optimal solution to $d_{haus}(f_\theta, \mathcal{F}_{TS}^\alpha)$ within a constant factor.*

*Proof.* The proof follows from Theorems 3.2 and 3.3, where we show that $d_{\text{haus}}$, $d_{\text{iso}}$, and triplet loss sampled by hard negative mining upper and lower bound each other by constant factors. Consequently, the optimal solution to triplet loss sampled by hard negative mining, and to $d_{\text{haus}}(f_\theta, \mathcal{F}_{TS}^\alpha)$, are equivalent within a constant factor. $\square$

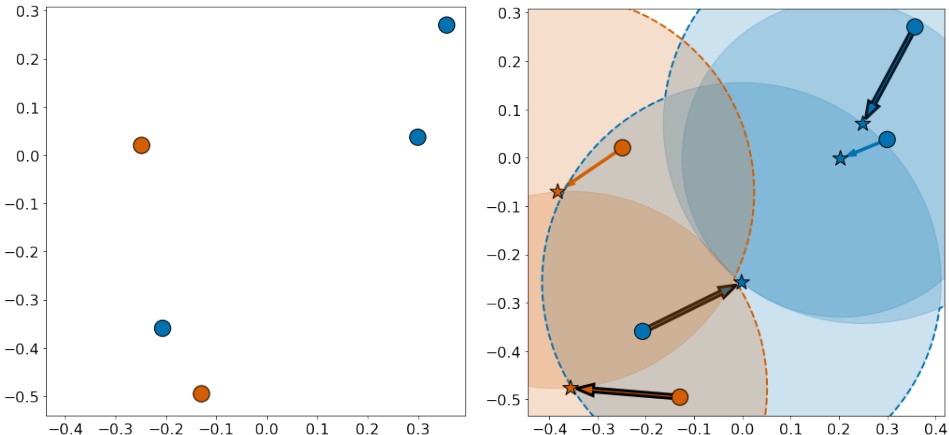

Figure 2: (Left) The setup for our toy example is a dataset of five arbitrary points, projected through $f_\theta$ onto the embedding space $\mathbb{R}^2$, divided into two classes, red and blue. (Right) Illustration of $d_{\text{haus}}(f_\theta, \mathcal{F}_{TS})$. The stars represent the embedding points of the function $f_{\text{haus}}^*$ that minimizes $d_{\text{haus}}(f_\theta, \mathcal{F}_{TS})$, and the arrows connect the points projected through $f_\theta$ and $f_{\text{haus}}^*$ respectively. The value of $d_{\text{haus}}(f_\theta, \mathcal{F}_{TS})$ is the maximum distance between two projections, whose arrows are outlined in black. The red and blue star sets are Triplet-Separated because they lie outside the other's Triplet-Separated boundary, indicated by the dashed colored border.

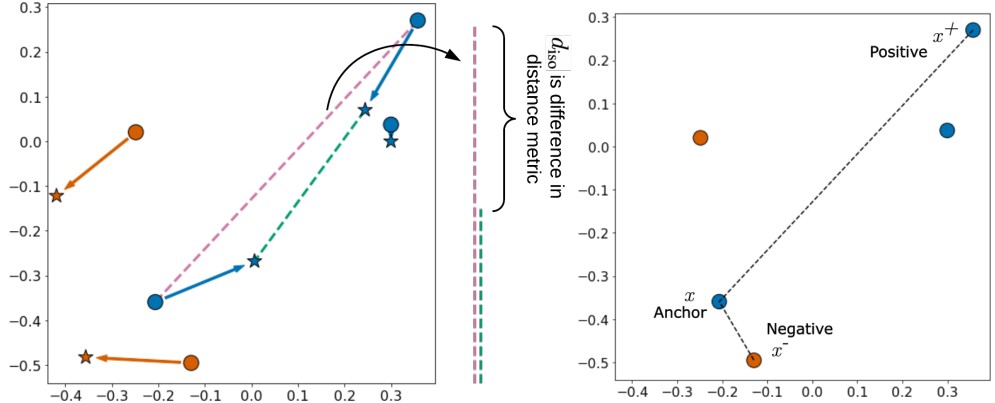

Figure 3: (Left) Illustration of $d_{\text{iso}}(f_\theta, \mathcal{F}_{TS})$. Using the toy example shown in Figure 2, we compute a $f_{\text{iso}}^*$ (marked in stars) that minimizes $d_{\text{iso}}(f_\theta, \mathcal{F}_{TS})$. To compute $d_{\text{iso}}(f_\theta, \mathcal{F}_{TS})$, we take the difference of the distance between two points under $f_\theta$ (circles) and the distance under $f_{\text{iso}}^*$ (stars) as shown by the curly brace in the middle. (Right) Illustration of the triplet loss sampled by hard negative mining. Using the toy example shown in Figure 2, we take the triplet (anchor, positive, negative) that maximizes the triplet loss.

### 3.2.4 ILLUSTRATIVE EXAMPLES FOR TRIPLET LOSS EQUIVALENCE

In this section, we illustrate the key ideas of the previous section's theorems by using a toy example with margin $\alpha = 0, N = 5$ points, and embedding dimension $d = 2$. As we will illustrate the equivalence between the triplet loss with the Hausdorff-like distance and isometric error, we can visualize the embedding points without any underlying data or neural network. See Figure 2 for the toy example setup.

Also shown in Figure 2 is a visualization of $d_{\text{haus}}(f_\theta, \mathcal{F}_{TS})$. The numerical value of $d_{\text{haus}}(f_\theta, \mathcal{F}_{TS})$ is determined by the maximum length of the arrows, which is marked in the figure with black outlines. Here, we compute the ideal $f_{\text{haus}}^*$, see (13), by optimizing the embedding points.

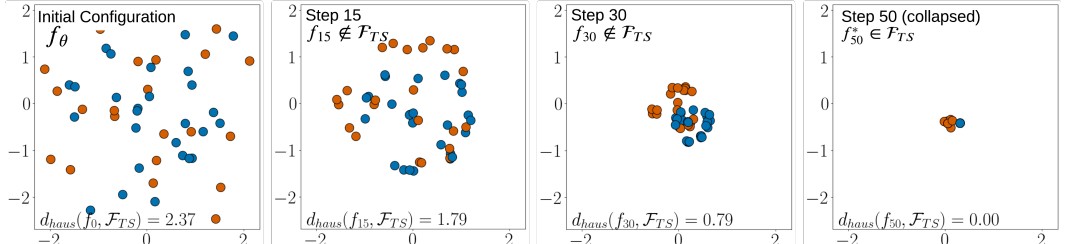

Figure 4: Illustration of how hard negative mining leads to collapsed solutions given $N = 20$ samples and embedding dimension $d = 2$. We approximate the $f^*$ that minimizes $d_{\text{haus}}(f_\theta, \mathcal{F}_{TS})$ by applying 50 gradient descent steps and observe that the embedding points collapse into a much smaller subset. Also marked are the values of $d_{\text{haus}}(f_i, \mathcal{F}_{TS})$ as measured by their Hausdorff-like distance from the approximated $f^*$.

Figure 3 illustrates $d_{\text{iso}}(f_\theta, \mathcal{F}_{TS})$, which measures the difference in distance metric. Note that the $f_{\text{iso}}^*$ that minimizes $d_{\text{iso}}(f_\theta, f_{\text{iso}}^*)$ is not necessarily the same as $f_{\text{haus}}^*$. Revisiting the second part of the proof for Theorem 3.2 (Also see Appendix A), $d_{\text{haus}}(f_\theta, \mathcal{F}_{TS})$ is lower bounded by $0.5 d_{\text{iso}}(f_\theta, \mathcal{F}_{TS})$ and upper bounded by $c d_{\text{iso}}(f_\theta, \mathcal{F}_{TS})$. For this specific toy example, we calculate the constant factor error to be $c = 0.53$. This essentially illustrates Theorem 3.2.

Lastly, we show the triplet loss sampled by hard negative mining on the right of Figure 3. The equivalence proved by Theorem 3.3 is shown by comparing the two figues in Figure 3, as the triplet selected by hard negative mining corresponds with the same three points with the largest discrepancies in distance metric. Through Figures 2 and 3, we have a visualization of the statement and proof of Corollary 3.4.

### 3.3 THEORETICAL INSIGHTS ON NETWORK COLLAPSE

We can use the embedding set diameter $\text{diam}(X_{f_\theta})$ to indicate for network collapse, as the diameter becomes near zero when the network is collapsed. Furthermore, if we assume that training always ends with a Triplet-Separated network $f_{TS}$ (collapsed or not), we have the following triangle inequality (16). The change in diameter cannot exceed twice the maximum displacement of each embedding point as measured by the Hausdorff-like distance (13).

$$|\text{diam}(X_{f_{TS}}) - \text{diam}(X_{f_{\text{init}}})| \leq 2 d_{\text{haus}}(f_{\text{init}}, \mathcal{F}_{TS}) \tag{16}$$

Re-arranging (16) and substituting in the constant-factor equivalence proven by Corollary 3.4, we find an inequality lower-bounding the diameter of the triplet-separated network.

$$\text{diam}(X_{f_{TS}}) \geq \text{diam}(X_{f_{\text{init}}}) - c \mathcal{L}_{\text{triplet}}(f_{\text{init}}) \tag{17}$$

From (17), we hypothesize two factors that influence network collapse are batch size and embedding dimension. As the batch size grows large, the triplet loss necessarily grows while the embedding diameter remains constant. Therefore we expect that large batch size leads to network collapse. On the other hand, increasing embedding dimension should increase the initial embedding diameter more than the triplet loss. As a result, low embedding dimension should lead to network collapse.

To illustrate our hypothesis that large batch size leads to network collapse, we show another toy example with $N = 20$ points with dimension $d = 2$ in Figure 4. The arrows illustrate the $f^*$ that minimizes $d_{\text{haus}}(f_\theta, \mathcal{F}_{TS})$ for this example, which is a collapsed function. We hypothesize that when $f^*$ is collapsed, the function $f_\theta$ learned by using the triplet loss would also be collapsed.

## 4 EXPERIMENTS

As mentioned in Section 1, current literature observes that hard negative mining results in network collapse inconsistently. Our theory proposes the testable hypothesis that network collapse happens when the batch size is too large, or when the embedding dimension is not large enough. We test

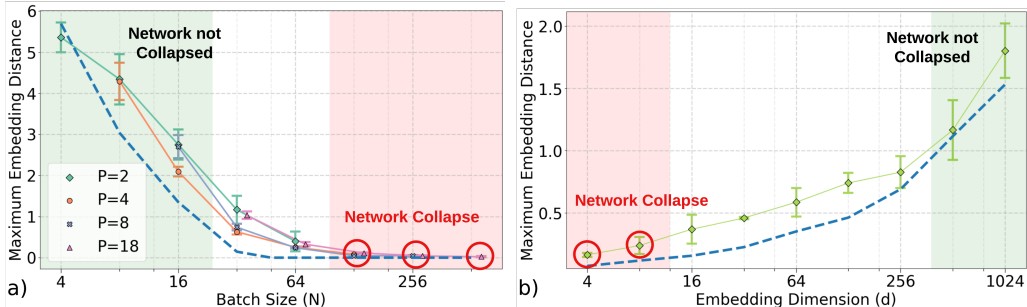

Figure 5: Experiments performed on the Market-1501 dataset with a randomly initialized ResNet-18 architecture and margin $\alpha = 0$. Each trial was repeated 3 times, and training is stopped on step 40,000. The theoretical diameter lower bound curve (17) for each experiment is drawn with dashed blue lines. (Left) Experiment A: Network embedding distances vary based on batch size with embedding dimension $d = 128$. As predicted by our theory, the embedding distances remain large for small batch sizes, with signs of collapse (shown in red) appearing when the batch size is large. The region marked in green indicates a lower probability of network collapse as corroborated by our experiment. (Right) Experiment B: Network embedding distances vary based on embedding dimension with batch size 32 ($P = 8, K = 4$). As predicted by our theory, the network embeddings are more likely to collapse when the embedding dimension is small and less when the embedding dimension is large.

our claims by first conducting an experiment on the Market1501 dataset Zheng et al. (2015) with a Resnet-18 He et al. (2016) architecture. This closely follows the work of Hermans et al. (2017), where the specific emphasis was on proving the efficacy of hard negative mining. To this end, we use a $PK$-style sampling method where $P$ people are sampled per batch and $K$ images are selected per person for a total batch size of $N = P \times K$, on the Market1501 dataset.

Our first experiment (see Figure 5 (a)), illustrates the effect of batch size on network collapse. Here, we use a fixed embedding dimension of $d = 128$, train until step 40,000, and repeat each trial 3 times. The batch size $P$ and $K$ are varied on a grid $P \in \{2, 4, 8, 18\}$ and $K \in \{2, 4, 8, 16, 32\}$ for a total of 20 combinations. Using (17) with constant value $c = 2$, we observe that the lower bound, shown in dashed blue, appears to hold across the tested batch size values. Further, as the batch size increases, we see that the embedding diameter decreases until there is a network collapse. This confirms our hypothesis that large batch size leads to network collapse.

We then conduct an additional experiment (see Figure 5 (b)) to study the relation between the embedding dimension and network collapse. Here, we first fix $P = 8$ and $K = 4$ for each batch and vary the embedding dimension $d$ from 4 to 1024. The network architecture and number of training steps are the same as the previous experiment (Figure 5 (a)). Once again, using (17) with $c = 2$, we observe that the theoretical diameter lower bound (dashed blue line) appears to hold across the tested embedding dimension values. Furthermore, as the embedding dimension decreases, we see that the embedding diameter also decreases to a point where the network collapses, thus confirming our second hypothesis.

It is worth noting that the experiments described above utilize a margin parameter $\alpha$ with a value of 0. While it is conjectured that the parameter $\alpha$ can be tuned to prevent network collapse, we show with an additional set of experiments that $\alpha$ does not play as significant a role in network collapse as the batch size or embedding dimension. Specifically, the experiments are conducted on the same Market-1501 dataset and Resnet-18 architecture but with $\alpha = 0.01$ and $\alpha = 0.05$. Larger $\alpha$ were also tested, but discarded due to significant overfit. These results are presented in Appendix C.

We further solidify our claims by conducting three more experiments on the Market1501 dataset Zheng et al. (2015) with a ResNet-50, a 2-layer convolutional network and GoogLeNet Szegedy et al. (2015) (see Appendix E for results). Branching out to other datasets, we also conducted three additional experiments on Stanford Online Products (SOP) Oh Song et al. (2016), Stanford Cars (CARS) Krause et al. (2013), and Caltech-UCSD Birds (CUB200) Wah et al. (2011) using a Resnet-18 architecture (see Appendix D for results). The same network collapse behavior has been

observed in each of these experiments, demonstrating the adverse effects of large batch size and low embedding dimension irrespective of dataset or network architecture.

In summary, we observe that the hard negative mining performs well with lower batch size, as reported by Hermans et al. (2017), and exhibits network collapse when the batch size is increased to a large value, which agrees with the triplet loss collapse reported by Schroff et al. (2015). This resolves the apparent contradiction from prior work and offers a unified explanation for network collapse in the context of hard negative mining.

## 5 DISCUSSION AND CONCLUSION

In this paper, we apply the isometric approximation theorem to prove that the triplet loss sampled by hard negative mining is equivalent to minimizing a Hausdorff-like distance. This mathematical foundation helps us present novel insights into hard negative mining by establishing a relationship between network collapse and the batch size or the embedding dimension. Our work presents mathematical proofs to support this relation and discusses extensive experiments to corroborate the same.

While we note that network collapse negatively affects the network's performance on downstream tasks, a lack of network collapse does not necessarily guarantee good performance. While simply choosing batch size to be small and embedding dimension to be large would be effective for avoiding network collapse, it is not necessarily good for optimizing network performance. Future work could involve looking at the effects of small batch size or large embedding dimension and recommending hyperparameter choices for batch size and embedding dimension.

Furthermore, it is worth noting that the isometric approximation theorem is independent of the triplet loss. Consequently, the theorem can be applied to any system utilizing the Euclidean metric, for instance, the pairwise contrastive loss Hadsell et al. (2006) ($\mathcal{L} = d(x, x^+) - [\alpha - d(x, x^-)]_+$) or the margin loss Wu et al. (2017) ($\mathcal{L} = [d(x, x^+) + \alpha - \beta]_+ + [-d(x, x^-) + \alpha + \beta]_+$). On the other hand, the unified metric learning formulation defined by Zheng et al. (2023) opens new avenues to explore alternative sampling methods and their respective functions in place of our Hausdorff-like metric $d_{\text{haus}}$. Through this and future work, we intend to leverage mathematical tools from functional analysis to explain fundamental principles in modern machine learning and artificial intelligence research.

## ACKNOWLEDGEMENTS

Thanks to Dr. Jeff Schneider for invaluable advice, discussion, and feedback. We would also like to thank the anonymouse reviewers for their comments and suggestions.

A portion of this research was sponsored by the Army Research Laboratory and was accomplished under Cooperative Agreement Number W911NF-20-2-0175. The views and conclusions contained in this document are those of the authors and should not be interpreted as representing the official policies, either expressed or implied, of the Army Research Laboratory or the U.S. Government. The U.S. Government is authorized to reproduce and distribute reprints for Government purposes notwithstanding any copyright notation herein.

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

## A    PROOF OF THEOREM 3.2

We now present Lemma A.1 that extends the results of Theorem 2.4 to apply to $d_{\text{iso}}$ and use this result to prove Theorem 3.2.

**Lemma A.1.** *If $d_{iso}(f,g) = \varepsilon$ and $\theta(f(\mathcal{X})) \geq q$, then there is a function $U$ isometric to $f$ such that:*

$$||g(x) - U(x)|| \leq c_n \varepsilon/q \ \forall x \in \mathcal{X} \tag{18}$$

*Proof.* If $f$ is invertible, then $gf^{-1}$ is a function $\mathbb{R}^n \to \mathbb{R}^n$. $gf^{-1}$ is an $\varepsilon$-nearisometry because $d_{\text{iso}}(f,g) = \varepsilon$. Then if $\theta(f(\mathcal{X})) \geq q$, the conditions for Theorem 2.4 are satisfied, so there exists an isometry $U_1 : \mathbb{R}^n \to \mathbb{R}^n$

$$||gf^{-1}(x) - U_1(x)|| \leq c_n \varepsilon/q \ \forall x \in f(\mathcal{X}) \tag{19}$$

Then

$$||g(x) - U_1(f(x))|| \leq c_n \varepsilon/q \ \forall x \in \mathcal{X} \tag{20}$$

Therefore if $f$ is invertible, (18) holds with $U = U_1 f$.

If $f$ is not invertible, then there exists $x_1 \neq x_2 \in \mathcal{X}$ such that $f(x_1) = f(x_2)$. We divide the elements of $\mathcal{X}$ into subsets $\mathcal{X}^\dagger$ and $\mathcal{X}'$ such that $f$ is invertible on $\mathcal{X}^\dagger$, $f(\mathcal{X}^\dagger) = f(\mathcal{X})$, and $d_{\text{iso}}$ is unchanged on $\mathcal{X}^\dagger$. Consequently, (20) holds on $\mathcal{X}^\dagger$.

Moving our attention to $\mathcal{X}'$, for all $x' \in \mathcal{X}'$ there exists $x^\dagger \in \mathcal{X}^\dagger$ such that $f(x') = f(x^\dagger)$. Then because $d_{\text{iso}}$ is unchanged on $\mathcal{X}^\dagger$, $||f(x') - g(x')|| \leq ||f(x^\dagger) - g(x^\dagger)|| \leq c_n \varepsilon/q$. Therefore (18) holds for $f$ and $g$ on $\mathcal{X}$. $\qquad\square$

*Proof.* [Theorem 3.2] We first prove that $d_{\text{iso}}$ upper bounds $d_{\text{haus}}$. To this end, fix the minimizing function $f_{\text{iso}}^*$ in the following expression:

$$d_{\text{iso}}(f_\theta, \mathcal{F}_{\alpha TS}) = \inf_{f_{\text{iso}}^* \in \mathcal{F}_{\alpha TS}} d_{\text{iso}}(f_\theta, f_{\text{iso}}^*) \tag{21}$$

From Lemma A.1 we have that:

$$\sup_{x \in \mathcal{X}} ||f_\theta(x) - f_{\text{iso}}^*(x)|| \leq c \, d_{\text{iso}}(f_\theta, f_{\text{iso}}^*) \tag{22}$$

with $c = c_n/\theta_*$. From the definition of Hausdorff-like distance (12) we have (23), and from (13) we have (24):

$$d_{\text{haus}}(f_\theta, f_{\text{iso}}^*) \leq \sup_{x \in \mathcal{X}} ||f_\theta(x) - f_{\text{iso}}^*(x)|| \tag{23}$$

$$d_{\text{haus}}(f_\theta, \mathcal{F}_{\alpha TS}) \leq d_{\text{haus}}(f_\theta, f_{\text{iso}}^*) \tag{24}$$

(25) follows from (22-24), proving that $d_{\text{iso}}$ upper bounds $d_{\text{haus}}$ within a constant factor of $c$.

$$d_{\text{haus}}(f_\theta, \mathcal{F}_{\alpha TS}) \leq c \, d_{\text{iso}}(f_\theta, \mathcal{F}_{\alpha TS}) \tag{25}$$

For the converse claim that $d_{\text{haus}}$ upper bounds $d_{\text{iso}}$, we once again fix the $f_{\text{haus}}^*$ that minimizes the following expression:

$$d_{\text{haus}}(f_\theta, \mathcal{F}_{\alpha TS}) = \sup_{x \in \mathcal{X}} ||f_\theta(x) - f_{\text{haus}}^*(x)|| \tag{26}$$

Next, for the four points $f_\theta(x_1)$, $f_\theta(x_2)$, $f_{\text{haus}}^*(x_1)$, and $f_{\text{haus}}^*(x_2)$, apply the triangle inequality via (27) to get (28).

$$||f_\theta(x_1) - f_\theta(x_2)|| \leq \begin{pmatrix} ||f_\theta(x_1) - f_{\text{haus}}^*(x_1)||+ \\ ||f_{\text{haus}}^*(x_1) - f_{\text{haus}}^*(x_2)||+ \\ ||f_{\text{haus}}^*(x_2) - f_\theta(x_2)|| \end{pmatrix} \tag{27}$$

$$\leq \begin{pmatrix} ||f_{\text{haus}}^*(x_1) - f_{\text{haus}}^*(x_2)||+ \\ 2 \sup_{x \in \mathcal{X}} ||f_\theta(x) - f_{\text{haus}}^*(x)|| \end{pmatrix} \tag{28}$$

It is worth noting that (28) holds for all $x_1, x_2 \in \mathcal{X}$. Furthermore, we can swap $f_\theta$ and $f^*_{\text{haus}}$ in (28) and use (26) to get (29) and thus (30).

$$d_{\text{iso}}(f_\theta, f^*_{\text{haus}}) = \tag{29}$$

$$\sup_{x_1, x_2 \in \mathcal{X}} \left| ||f_\theta(x_1) - f_\theta(x_2)|| - ||f^*_{\text{haus}}(x_1) - f^*_{\text{haus}}(x_2)|| \right| \leq$$

$$2 d_{\text{haus}}(f_\theta, \mathcal{F}_{\alpha TS})$$

$$d_{\text{iso}}(f_\theta, \mathcal{F}_{\alpha TS}) \leq d_{\text{iso}}(f_\theta, f^*_{\text{haus}}) \leq 2 d_{\text{haus}}(f_\theta, \mathcal{F}_{\alpha TS}) \tag{30}$$

(30) proves that $d_{\text{haus}}$ upper bounds $d_{\text{iso}}$ within a constant factor of 2.

Thus we prove that $d_{\text{haus}}$ and $d_{\text{iso}}$ upper bound each other within constant factors. $\qquad\square$

## B  PROOF OF THEOREM 3.3

*Proof.* [Theorem 3.3] Here, we present a detailed proof for the theorem using equations (31-42)

From the definition of $d_{\text{iso}}$ in (31), we introduce the anchor, positive, and negative triplet $(x, x^+, x^-)$ in (32) by re-labelling $x_1 \to x$. Recognizing that $x_2$ must either have the same or different label from $x_1$, we re-label $x_2 \to x^+$ or $x_2 \to x^-$, and pick the max of these distances for any given triplet.

$$d_{\text{iso}}(f_\theta, \mathcal{F}_{\alpha TS}) = \inf_{f^* \in \mathcal{F}_{\alpha TS}} \sup_{x_1, x_2 \in \mathcal{X}} \left| ||f_\theta(x_1) - f_\theta(x_2)|| - ||f^*(x_1) - f^*(x_2)|| \right| \tag{31}$$

$$= \inf_{f^* \in \mathcal{F}_{\alpha TS}} \sup_{x, x^+, x^-} \max \left\{ \left| ||f_\theta(x) - f_\theta(x^+)|| - ||f^*(x) - f^*(x^+)|| \right|, \left| ||f_\theta(x) - f_\theta(x^-)|| - ||f^*(x) - f^*(x^-)|| \right| \right\} \tag{32}$$

Inequality (33) follows from $\max(a, b) \leq a + b$ for positive $a, b$.

$$\leq \inf_{f^* \in \mathcal{F}_{\alpha TS}} \sup_{x, x^+, x^-} \left| ||f_\theta(x) - f_\theta(x^+)|| - ||f^*(x) - f^*(x^+)|| \right| + \left| ||f_\theta(x) - f_\theta(x^-)|| - ||f^*(x) - f^*(x^-)|| \right| \tag{33}$$

Now fix the $f^*$ that minimizes (33). We next prove via contradiction that the first term (34a) is positive and the second term (34b) is negative.

$$||f_\theta(x) - f_\theta(x^+)|| - ||f^*(x) - f^*(x^+)|| \tag{34a}$$

$$||f_\theta(x) - f_\theta(x^-)|| - ||f^*(x) - f^*(x^-)|| \tag{34b}$$

There are four cases we must consider here, as we treat the zero case as either positive or negative. Case 1: (34a) is positive, (34b) is positive. Denoting this as $++$, our four cases are $(1 : ++)$, $(2 : --)$, $(3 : -+)$, $(4 : +-)$. Now we prove by contradiction that case 4 is the only valid one.

Case 1($++$): Consider the function $f^\dagger(x) = (1 + \delta) f^*(x)$ where $\delta > 0$ is a small constant. Then $d_{\text{iso}}(f_\theta, f^\dagger) < d_{\text{iso}}(f_\theta, f^*)$, contradicting the statement that $f^*$ minimizes $d_{\text{iso}}$.

Case 2($--$): Consider the function $f^\dagger(x) = (1 - \delta) f^*(x)$ where $\delta > 0$ is a small constant. Then $d_{\text{iso}}(f_\theta, f^\dagger) < d_{\text{iso}}(f_\theta, f^*)$, contradicting the statement that $f^*$ minimizes $d_{\text{iso}}$.

Case 3($-+$): We can algebraically rearrange (33) to get:

$$||f^*(x) - f^*(x^+)|| - ||f^*(x) - f^*(x^-)|| - ||f_\theta(x) - f_\theta(x^+)|| + ||f_\theta(x) - f_\theta(x^-)|| \geq 0 \tag{35}$$

$$||f^*(x) - f^*(x^+)|| - ||f^*(x) - f^*(x^-)|| \leq 0 \tag{36}$$

$$-\left( ||f_\theta(x) - f_\theta(x^+)|| - ||f_\theta(x) - f_\theta(x^-)|| \right) \geq 0 \tag{37}$$

(36) comes from the definition of $f^*$ as a Triplet-Separated function; then (37) comes from combining (35) and (36). However, this means that the triplet that maximizes the expression has negative triplet loss, therefore there must be some other $f_2^*$ with a smaller value. This contradicts the statement that $f^*$ minimizes $d_{\text{iso}}$.

With Cases 1, 2, and 3 eliminated, we only have Case 4 and all the zero cases $(00, +0, -0, 0+, 0-)$. We note that the cases $-0$ and $0+$ can be dis-proven using the same logic as Case 3. This leaves the four following valid cases $(00, 0-, +0, +-)$, where we can connect back to (33) and write:

$$\sup_{x,x^+,x^-} \left| ||f_\theta(x) - f_\theta(x^+)|| - ||f^*(x) - f^*(x^+)|| \right| + \left| ||f_\theta(x) - f_\theta(x^-)|| - ||f^*(x) - f^*(x^-)|| \right| \tag{38}$$

$$= \sup_{x,x^+,x^-} ||f_\theta(x) - f_\theta(x^+)|| - ||f_\theta(x) - f_\theta(x^-)|| - \left( ||f^*(x) - f^*(x^+)|| - ||f^*(x) - f^*(x^-)|| \right) \tag{39}$$

Note that (39) resembles the triplet loss. The triplet loss for $f^*$ cannot dominate the maximum triplet loss for $f_\theta$, otherwise it would contradict the statement that $f^*$ minimizes the isometric error, giving us:

$$-\left( ||f^*(x) - f^*(x^+)|| - ||f^*(x) - f^*(x^-)|| \right) \le \sup_{x,x^+,x^-} ||f_\theta(x) - f_\theta(x^+)|| - ||f_\theta(x) - f_\theta(x^-)|| \tag{40}$$

$$\tag{41}$$

Using (40), we have the following relation with respect to (39).

$$\le 2 \sup_{x,x^+,x^-} ||f_\theta(x) - f_\theta(x^+)|| - ||f_\theta(x) - f_\theta(x^-)|| + \alpha \tag{42}$$

Note that (42) is identical to twice the expression for the triplet loss sampled by hard negative mining. Therefore the triplet loss sampled by hard negative mining upper bounds the isometric error by a constant factor of 2.

Additionally, we can prove that the triplet loss sampled by hard negative mining upper bounds the isometric error. Starting from the definition of isometric error in (43), inequality (44) follows from $\max(a, b) \ge (a + b)/2$.

$$d_{\text{iso}}(f_\theta, \mathcal{F}_{\alpha TS}) = \inf_{f^* \in \mathcal{F}_{\alpha TS}} \sup_{x_1, x_2 \in \mathcal{X}} \left| ||f_\theta(x_1) - f_\theta(x_2)|| - ||f^*(x_1) - f^*(x_2)|| \right| \tag{43}$$

$$\ge \frac{1}{2} \inf_{f^* \in \mathcal{F}_{\alpha TS}} \sup_{x,x^+,x^-} \left| ||f_\theta(x) - f_\theta(x^+)|| - ||f^*(x) - f^*(x^+)|| \right| + \left| ||f_\theta(x) - f_\theta(x^-)|| - ||f^*(x) - f^*(x^-)|| \right| \tag{44}$$

Once again fixing $f^*$, we have equality (45) by the same logic as the previous part. Inequality (46) follows from the fact that $||f^*(x) - f^*(x^+)|| - ||f^*(x) - f^*(x^-)|| \le -\alpha$ by the definition of $f^*$ as $\alpha$ Triplet-Separated.

$$= \frac{1}{2} \sup_{x,x^+,x^-} ||f_\theta(x) - f_\theta(x^+)|| - ||f_\theta(x) - f_\theta(x^-)|| - \left( ||f^*(x) - f^*(x^+)|| - ||f^*(x) - f^*(x^-)|| \right) \tag{45}$$

$$\ge \frac{1}{2} \sup_{x,x^+,x^-} ||f_\theta(x) - f_\theta(x^+)|| - ||f_\theta(x) - f_\theta(x^-)|| + \alpha \tag{46}$$

Therefore isometric error upper bounds the triplet loss sampled by hard negative mining by a constant factor of 2. □

## C    ADDITIONAL EXPERIMENTS FOR NON-ZERO $\alpha$

In the experiments shown in the main paper, we fixed $\alpha$ to be zero. To prove that using a nonzero $\alpha$ does not affect the results, we repeated the Market1501 experiments with $\alpha = 0.01$ and $0.05$, observing that our results remain consistent. See Figure 6 for the embedding diameter vs batch size plots, which follow the same trends as the $\alpha = 0$ experiment shown in Figure 5. Similarly, the embedding diameter vs dimension plots in Figure 7 also follow similar trends to those shown by the $\alpha = 0$ experiments in Figure 5.

In addition, we also conducted experiments with a larger $\alpha = 0.1$, but decided to stop there as the trained networks began to exhibit significant overfit. In particular, the validation loss values began to increase while the training loss flatlined at the $\mathcal{L} = 0.1$ line. As a result, the final embedding diameters on the validation data did not show any meaningful pattern, so we choose not to show those plots in the appendix.

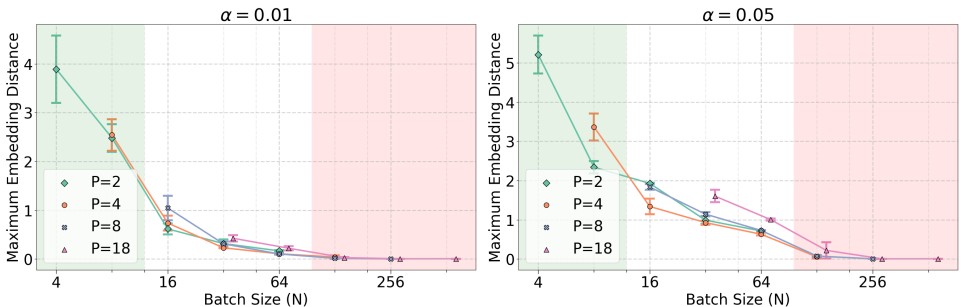

Figure 6: Embedding diameter vs batch size experiments with non-zero $\alpha$ on Market1501 dataset for dimension $d = 128$. The left figure ($\alpha = 0.01$) and right figure ($\alpha = 0.05$) both show network collapse when the batch size is large (red region), and do not collapse when the batch size is smaller (green region). We note that the embedding diameters are larger when $\alpha$ is larger, which is somewhat to be expected. However there appears to be a collapse threshold around $N = 128$ independent of $\alpha$ which may warrant further study in a future work.

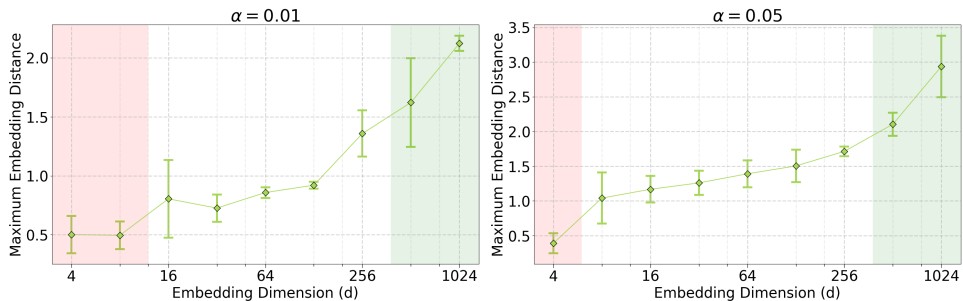

Figure 7: Embedding diameter vs dimension experiments with non-zero $\alpha$ on Market1501 dataset for batch size 32 ($P = 8, K = 4$). The left ($\alpha = 0.01$) and right ($\alpha = 0.05$) both show signs of network collapse (red region) for lower dimension and larger embedding diameter (green region) for higher dimension, mirroring the $\alpha = 0$ experiment observations.

## D    ADDITIONAL EXPERIMENTS FOR CUB200, SOP, AND CARS DATASETS

In the main paper, our experiments were conducted only on the Market1501 dataset. To show that our results generalize beyond just one dataset, we have replicated the same experiments on 3 additional datasets: Stanford Online Products (SOP) Oh Song et al. (2016), Caltech Birds (CUB200) Wah et al. (2011), and Stanford Cars (CARS) Krause et al. (2013). We train a Resnet18 network from scratch for each experiment with 3 repetitions for each hyperparameter combination, and leave the margin parameter $\alpha = 0$. One difference from the main experiments is that these three datasets

do not lend themselves to the $PK$-style sampling that Market1501 uses, so we use a fixed batch size without the same-class guarantees that exist for $PK$-style sampling. Results are shown in Figures 8 and 9.

On the SOP dataset (Figure 8), we observe that networks collapse for large batch size and small embedding dimension, exactly the same as the Market-1501 dataset shown in the main paper. For the CUB and CARS datasets (Figure 9), we also observe that networks collapse for large batch size. However, when experimenting with the embedding dimension, we observed that the CUB and CARS datasets are extremely sensitive to the batch size; a lower batch size would never collapse the network, and a higher batch size would always collapse the network. To mitigate this sensitivity, we attempted to explore larger models with higher embedding dimensions than 1024, but were constrained by our hardware and GPU specifications.

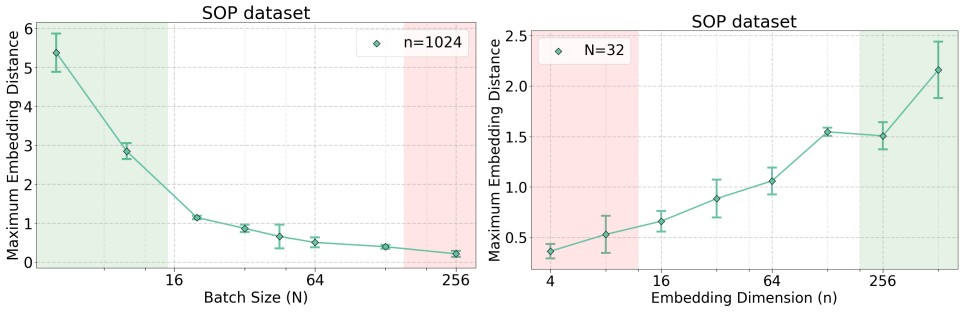

Figure 8: Batch size and embedding dimension experiments on SOP dataset using Resnet18 and $\alpha = 0$. We observe that the patterns of collapse (red region) for high batch size and low embedding dimension hold for the SOP dataset.

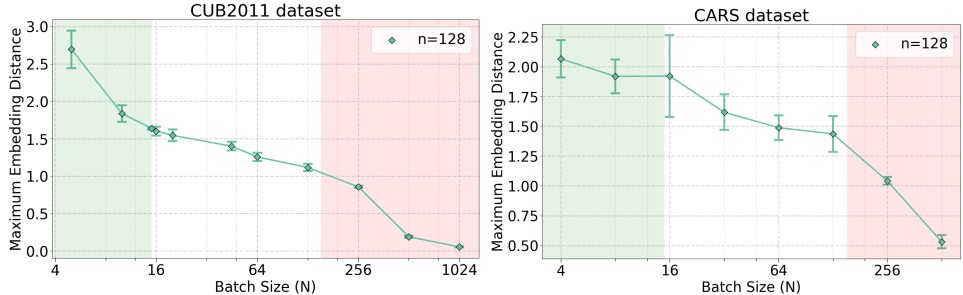

Figure 9: Batch size experiments on CUB200 and CARS dataset using Resnet18 and $\alpha = 0$. We observe that the patterns of collapse (red region) for high batch size hold for both datasets.

# E   ADDITIONAL EXPERIMENTS WITH DIFFERENT BASE NETWORK

As our theory only assumes that an ideal $\alpha$-Triplet Separated network exists in the function space parameterized by the neural net's architecture, it stands to reason that our theory should be somewhat architecture-agnostic. In the main paper, we used a Resnet-18 base, and in this appendix we train a randomly initialized ResNet-50 architecture, GoogLeNet architecture Szegedy et al. (2015) as well as a simple 2-layer convolutional network. The results for ResNet-50 are shown in Figure 10. We do observe that the networks collapse for large batch size and small embedding dimension, just as they do using the ResNet-18 backbone.

The results for GoogLeNet are shown in Figure 11. We do observe that the networks collapse for large batch size and small embedding dimension, just as they do using the ResNet-18 backbone. We also observe a few instances of network collapse for a high embedding dimension. However, we noted memory warnings during training due to the large size of the network, and we attribute these anomalies to our hardware constraints.

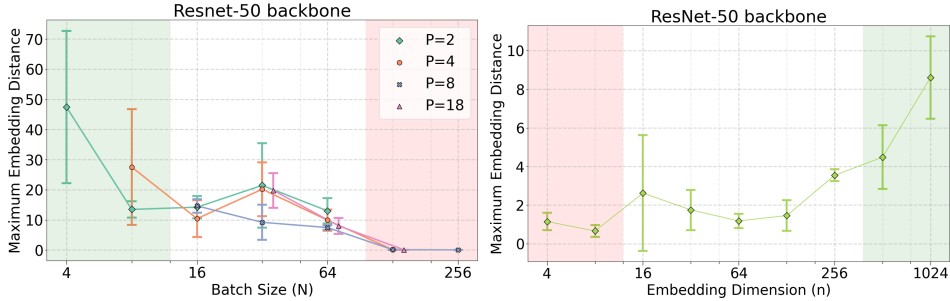

Figure 10: Batch size and embedding dimension experiments on Market1501 dataset using a ResNet-50 network and $\alpha = 0$. We observe that patterns of collapse (red region) for high batch size and low embedding dimension hold for a ResNet-50 network, just as they do for Resnet-18 in Figure 5 of Section 4.

However, we also observed a couple instances of collapse for high embedding dimension, where our theory hypothesized that collapse would not occur. We note that while training the high embedding dimension networks, our code was raising memory warnings because the network was too large, so we suspect that these results may not be entirely accurate.

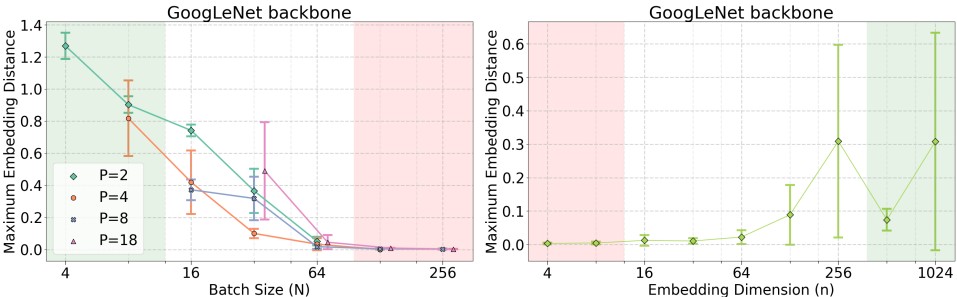

Figure 11: Batch size and embedding dimension experiments on Market1501 dataset using GoogLeNet and $\alpha = 0$. We observe that the patterns of collapse (red region) for high batch size and low embedding dimension hold for a GoogLeNet architecture, just as they do for Resnet-18 in the main paper. We note that the embedding dimension experiments have a few outliers for $d > 256$. We suspect our code ran into memory issues.

On the other hand, the results for a 2-layer convolutional network are shown in Figure 12. Like with GoogLeNet and ResNet, we observe that large batch size and small embedding dimension both lead to network collapse.

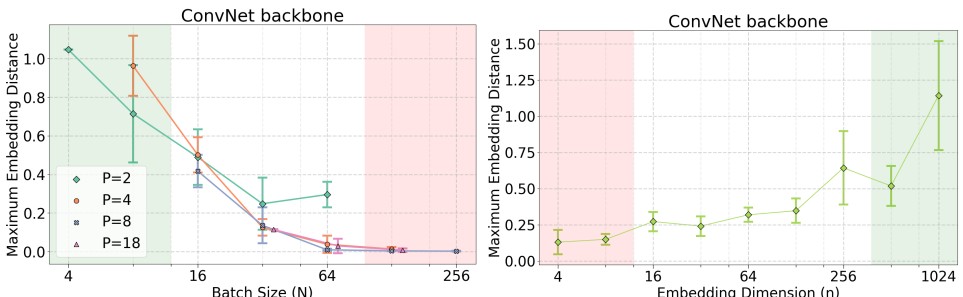

Figure 12: Batch size and embedding dimension experiments on Market1501 dataset using a 2-layer Convolutional network and $\alpha = 0$. We observe that patterns of collapse (red region) for high batch size and low embedding dimension hold for a 2-layer convolutional network, just as they do for Resnet-18 in the main paper.

