# OpenReview forum: "Mathematical Justification of Hard Negative Mining via Isometric Approximation Theorem"
_ICLR.cc/2024/Conference — ICLR 2024 poster_

### Official Review · Reviewer_DHxZ · 2023-10-25

**Soundness:** 3 good
**Presentation:** 3 good
**Contribution:** 3 good
**Rating:** 8
**Confidence:** 5

**Summary:**

This theoretical paper discusses and analyzes the collapse problem in the Deep Metric Learning area. The author leverages the theory of isometric approximation to show an equivalence between the triplet loss sampled by hard negative mining and an optimization problem that minimizes a Hausdorff-like distance between the neural network and its ideal counterpart function. The main conclusion is equation 17, indicating that network collapse tends to happen when batch size is too large or the embedding dimension is too small. Effective experiments were conducted to visualize and support the conclusion.

**Strengths:**

The collapse problem is a famous problem blocking DML training. Current literature misses the systematic explanation of this phenomenon. The theory of the paper is helping understand the reason behind the problem and will benefit the DML research community.

The starting point of the definition of triplet-separated accurately describes and fits the optimization of triplet loss.
By leveraging the isometric approximation theorem, the path to achieve the proof to the corollary 3.4 is clear and sound.
Finally, the experiments with different settings to proof the conclusion is equation 17 is wee conducted and integrated

**Weaknesses:**

The illustrative examples are not clear. Figure 2 is clear on the examples. But the figure 3 is not clear.
1. The line in figure 3 overlaps too much. would be good to redraw it.
2. it is not clear of diso(fθ,FTS) in the figure.

Figure 4 can be extended to multiple views to illustrate the progress of the collapse as the example shown in [1]. and if the diso(fθ,FTS) value can be added in the vis, it will be valued.

miner issue in the typos: section 1 "CUBwah et al. (2011))"

[1]: Hong Xuan, Abby Stylianou, Xiaotong Liu, and Robert Pless. Hard negative examples are hard but useful.InEuropeanConferenceonComputerVision, pp.126–142.Springer,2020.

**Questions:**

Some extended questions:
How is the lower bound curve (17) related to the retrieval performance?
Triplet loss is rarely used in Self-supervised learning with a random initialization network. Maybe one of the reasons is also collapse. Can your theory explain the problem? is the DML problem similar to the SSL problem in the embedding level?

---

> ### Author Response · Authors · 2023-11-17
>
> We thank the reviewer for the constructive comments. We respond to the comments below and update the manuscript accordingly.
>
>
> **$\mathbf Q_1$: "...figure 3 is not clear"**
>
> $A_1$: Thank you for the feedback. We have updated the figure to remove extraneous lines.
> The difference in the distance metric should now be more evident.
> Further, we improved the clarity of the object marked by $d_\textrm{iso}(f_\theta,\mathcal F_{TS})$ by removing the clutter of lines surrounding the text in the figure.
> We hope this addresses the illustration in Fig. 3.
>
>
> **$\mathbf Q_2$: "Figure 4 can be extended to multiple views to illustrate..."**
>
> $A_2$: Thank you for the suggestion and for the reference. We have modified our Fig. 4 to match the style in [1]. Additionally, we compute $d_\textrm{haus}(f_\theta,\mathcal F_{TS})$ as $d_\textrm{haus}(f_\theta, f^*)$ using the approximated $f^*$ function and have marked the function value in Fig. 4 to provide additional context.
>
>
> **$\mathbf Q_3$: "typos: section 1..."**
>
> $A_3$: Thank you for pointing this out. We have fixed the typo in the revised version.
>
> **$\mathbf Q_4$: "How is the lower bound curve (17) related to the retrieval performance?"**
>
> $A_4$: We are unaware of any strong theoretical connection between (17) and retrieval performance.
> There is no guarantee that retrieval performance will always improve as the diameter increases.
> However, in the converse, we know retrieval performance will be negatively impacted when the diameter goes to zero, indicating network collapse.
>
> **$\mathbf Q_5$: "Triplet loss is rarely used in Self-supervised learning... is the DML problem similar to the SSL problem in the embedding level?"**
>
> $A_5$: The reviewer points to an interesting connection between self-supervised learning (SSL) and deep metric learning (DML) at the embedding level. While we acknowledge a similarity in how SSL and DML use contrastive loss, there exists a difference in the structures of their respective datasets.
> We believe exploring and presenting the extension of our theory to SSL will likely need new terminology and careful consideration in the proofs, and thus we keep this for future work.

---

### Official Review · Reviewer_hpyY · 2023-11-02

**Soundness:** 1 poor
**Presentation:** 2 fair
**Contribution:** 2 fair
**Rating:** 3
**Confidence:** 4

**Summary:**

The paper focuses on addressing the issue of network collapse in deep metric learning, specifically in the context of the triplet loss. Network collapse refers to a situation where the network projects all data points onto a single point, leading to ineffective embeddings. The authors propose utilizing the mathematical theory of isometric approximation to establish an equivalence between the triplet loss with hard negative mining and an optimization problem that minimizes a Hausdorff-like distance. This theoretical framework provides the justification for the empirical success of hard negative mining in preventing network collapse. Experimental results on multiple datasets and network architectures validate the theoretical findings.

**Strengths:**

-	The paper offers a theoretical framework based on isometric approximation theory to explain the behavior of the triplet loss with hard negative mining. This provides a solid foundation for understanding and addressing network collapse in deep metric learning
-	The paper investigates the factors that contribute to network collapse, specifically focusing on the batch size and embedding dimension. Through experiments on multiple datasets and network architectures, the authors demonstrate that larger batch sizes and smaller embedding dimensions increase the likelihood of network collapse.
-	The authors conduct experiments on various datasets and network architectures, demonstrating the effectiveness of hard negative mining in preventing network collapse. The empirical results corroborate the theoretical findings and enhance the credibility of the proposed approach.

**Weaknesses:**

-	The paper primarily focuses on hard negative mining as a solution to network collapse but does not thoroughly compare its performance with alternative methods or strategies. A comprehensive comparison with other approaches would provide a more comprehensive understanding of the effectiveness of hard negative mining and its relative advantages or disadvantages compared to other techniques. Without such comparisons, it is difficult to assess the competitiveness of the proposed method.
-	While the paper presents experimental results on multiple datasets and network architectures, it does not provide precise quantitative measurements of the performance improvements achieved by preventing network collapse through hard negative mining. Without specific metrics, such as accuracy or loss values, it is challenging to assess the magnitude of the improvement or compare it with alternative approaches. Including quantitative evaluations would enhance the rigor and credibility of the proposed method.
-	The absence of mining time analysis and discussion on the time complexity limits the applicability of the proposed method to real-time or time-sensitive applications. Real-time systems often require fast and efficient processing, and the computational cost of hard negative mining can be a critical factor in determining the feasibility of the approach. Considering the practical implications and performance trade-offs in real-time scenarios would enhance the relevance and applicability of the proposed method.
-	The paper primarily focuses on the theoretical framework and experimental results of hard negative mining in preventing network collapse. However, the experiments are conducted on a limited set of network architectures, and it does not include widely used backbone architectures such as ResNet-50 or Inception-BN. These mainstream architectures are commonly employed in deep metric learning and computer vision tasks. The lack of evaluation on such architectures limits the generalizability and applicability of the proposed method to real-world scenarios.

**Questions:**

See Weaknesses

---

> ### Author Response · Authors · 2023-11-18
>
> We thank the reviewer for the constructive comments. We respond to the comments below and update the manuscript accordingly.
>
> **$\mathbf Q_1$: "...comparison with other approaches would provide a more comprehensive understanding..."**
>
> $A_1$: Thank you for the question. As we state in the fifth paragraph of section 1, this work aims to explain why network collapse happens by using the theory of isometric approximation to draw an equivalence between the triplet loss with hard negative mining and Hausdorff-like distance metric. We would like to clarify that our paper does not propose a new method, and the performance of the triplet loss with hard negative mining has already been recorded in other papers.
>
> Prior works such as [1] point to a network collapse in the context of hard negative mining while the authors in [2] show the absence of network collapse with hard negative mining for a similar dataset. To address these seemingly contradictory results, our work offers a systematic explanation for network collapse, its causes, and how to avoid it in the context of triplet loss with hard negative mining.
>
> [1] Florian Schroff, Dmitry Kalenichenko, and James Philbin. Facenet: A unified embedding for face
> recognition and clustering. In Proceedings of the IEEE conference on computer vision and pattern
> recognition, pp. 815–823, 2015.
>
> [2] Alexander Hermans, Lucas Beyer, and Bastian Leibe. In defense of the triplet loss for person reidentification.
> arXiv preprint arXiv:1703.07737, 2017.
>
> **$\mathbf Q_2$: "Including quantitative evaluations would enhance the rigor and credibility of the proposed method."**
>
> $A_2$: Continuing our response from the previous section, the focus of this work is to provide a mathematical justification for network collapse and not present a new method to perform hard negative mining. Consequently, our experiments are directed to corroborate the likelihood of network collapse when the batch size is large or when the embedding dimension is small and not to provide precise quantitative measurements for hard negative mining.
>
> **$\mathbf Q_3$: "The absence of mining time analysis and discussion on the time complexity..."**
>
> $A_3$: We feel that the mining time analysis is not relevant to this work. While it is true that discussion on time complexity is an essential factor when proposing new triplet sampling methods, our work primarily investigates the phenomenon of network collapse on an existing method, i.e., hard negative mining.
>
> **$\mathbf Q_4$: "...experiments are conducted on a limited set of network architectures..."**
>
> $A_4$: While our initial submission reports findings from three different network architectures (ResNet-18, GoogLeNet, and a 2-layer convnet), we have now conducted experiments with ResNet-50 as the backbone, as per the reviewer's suggestion. The results from this new experiment corroborate our claims that network collapse is more likely when the batch size is large or when the embedding dimension is small. We have now included the details of these results in Appendix E and highlighted the changes in yellow. We thank the reviewer for the suggestion, since including these additional results has surely strengthened our manuscript.

---

### Official Review · Reviewer_Mw75 · 2023-11-07

**Soundness:** 2 fair
**Presentation:** 2 fair
**Contribution:** 2 fair
**Rating:** 5
**Confidence:** 3

**Summary:**

This paper delves into the issue of triplet loss collapse, offering a theoretical insight into its causes. The proposed method employs isometric approximation to demonstrate that hard example mining is effectively equivalent to minimizing a distance akin to the Hausdorff distance. Experimental results on two widely used metric learning datasets reveal that network collapse tends to occur with either a large batch size or a small embedding size.

**Strengths:**

- This paper addresses a significant issue related to triplet loss collapse, a topic of keen interest within the research community.

- The paper offers a novel theoretical perspective on the problem of triplet loss collapse by introducing the innovative concept of isometric approximation, contributing a fresh angle to this field.

**Weaknesses:**

- It would be beneficial to see a more thorough analysis of various triplet loss variants, particularly with regard to different triplet sampling methods. This in-depth examination is essential for a comprehensive understanding of the effectiveness of triplet loss in different settings.

- The figures and examples in this paper suffer from both low quality and a lack of informativeness. They do not effectively convey the discussed settings, making it difficult for readers to comprehend the content.

- The experimental evaluations in this paper are notably limited, making it challenging to substantiate the claims put forth. For instance, the use of a very large batch size with semi-hard triplets in FaceNet raises questions that require further clarification or justification.

**Questions:**

See weakness.

---

> ### Author Response · Authors · 2023-11-17
>
> We thank the reviewer for the constructive comments. We respond to the comments below and update the manuscript accordingly.
>
> **$\mathbf Q_1$: "...a more thorough analysis of various triplet loss variants..."**
>
> $A_1$: While our work focuses on hard negative mining on the triplet loss, it is worth noting that the theory of isometric approximation as used in this manuscript is independent of the triplet loss. Consequently, it can be applied to any system utilizing the Euclidean metric, for instance the pairwise contrastive loss ($\mathcal{L} = |x - x^+| - [\alpha - |x-x^-|]_+$) or the margin loss ($\mathcal{L}=\left[|x-x^+| + \alpha - \beta\right]\_+ + [-|x-x^-| + \alpha + \beta]\_+$).
>
> On the other hand, the fundamental reason for the choice of hard negative mining is that it resembles a $\max_{x,x^+,x^-} \mathcal L(x, x^+, x^-)$ type of operation, which via the isometric approximation theorem, can be directly connected with our defined Hausdorff-like metric $d_\textrm{haus}(f_1, f_2) = \max_x d(f_1(x), X_{f_2}^i)$ or $\max_x d(f_2(x), X_{f_1}^i)$, because they both use max operations.
> To rigorously examine other triplet sampling methods, we would likely need a careful exploration of functions that naturally connect with those other sampling methods, and thus we keep this for future work.
>
> **$\mathbf Q_2$: "The figures and examples in this paper suffer from both low quality and a lack of informativeness..."**
>
> $A_2$: Thank you for the feedback. We have clarified the caption of Fig. 2 to better describe the example toy system.
>
> We have removed extraneous lines from Fig. 3, which now exclusively focuses on a single example for illustrating $d_\textrm{iso}$.
> This should remove any confusion from multiple examples in our prior version and should make the difference in the distance metric more evident. Further, we improved the clarity of the object marked by $d_\textrm{iso}(f_\theta,\mathcal F_{TS})$ by removing the clutter of lines surrounding the text in the figure. We hope this improves the clarity around Fig. 3.
>
> With respect to Fig. 4, we have added a sequential collapse of a solution with insights into how the function looks at intermediate steps. We have also added annotations illustrating the decrease of $d_\textrm{haus}(f_\theta, \mathcal F_{TS})$ for their respective functions. We believe this provides additional context to our figures and examples. If there is some other detail that the reviewer is pointing to, we would be happy to address that as well.
>
> **$\mathbf Q_3$: "The experimental evaluations in this paper are notably limited, making it challenging to substantiate the claims put forth. For instance, the use of a very large batch size with semi-hard triplets in FaceNet raises questions that require further clarification or justification."**
>
> $A_3$: Thank you for the feedback. Quoting from the FaceNet [1] paper, ``Selecting the hardest negatives can in practice lead to bad local minima early on in training, specifically it can result in a collapsed model (i.e. f(x) = 0).'' FaceNet [1] work uses this fact as a motivating example to propose semi-hard triplets to work with very large batch sizes.
> Our work supports and justifies their observations that very large batch sizes in combination with hard negative mining adversely affects the network.
>
> In this work, we provide rigorous mathematical analysis to show how large batch sizes and small embedding dimensions adversely affect hard negative mining. Additionally, we present experimental verification with the person re-identification dataset (Market-1501) reconcile the findings of FaceNet [1], where a batch size in the order of thousands led to network collapse, and [2] where a batch size of N = 72 showed no network collapse. We
> further support our predictions via experiments spanning three additional datasets (SOP, CARS, and CUB200) and three different network
> architectures (ResNet-18, GoogLeNet, and a 2-layer
> convnet). We are happy to conduct additional experiments on any specific datasets or network architectures to expand the breadth of our verification, as per the reviewer's additional requests.
>
> Furthermore, our recent experiments with ResNet-50 as the backbone resonate with the findings from the previous three network architectures. We hope the experimental evaluation is now adequate to substantiate the claims put forth in the manuscript.
> We are happy to conduct supplementary experiments on specific datasets or network architectures to expand the breadth of our verification per the reviewer's additional requests.
>
> [1] Florian Schroff, Dmitry Kalenichenko, and James Philbin. Facenet: A unified embedding for face
> recognition and clustering. In Proceedings of the IEEE conference on computer vision and pattern
> recognition, pp. 815–823, 2015.
>
> [2] Alexander Hermans, Lucas Beyer, and Bastian Leibe. In defense of the triplet loss for person reidentification.
> arXiv preprint arXiv:1703.07737, 2017.

---

### Official Review · Reviewer_QYwJ · 2023-11-09

**Soundness:** 3 good
**Presentation:** 4 excellent
**Contribution:** 2 fair
**Rating:** 6
**Confidence:** 4

**Summary:**

In this paper, the authors attempt to analyse the network collapse phenomenon associated with hard negative mining for triplet loss. Such network collapse is not omnipresent and vary depending on the task at hand. The authors attempt to explain this phenomena by drawing a equivalence between the hard negative-mining triplet loss and the Hausdorff-like distance metric by using isometric approximation. Such approximations help to establish a relationship between the network collapse and the batch size or the dimension of the embedding, which provide insights into using different batch size or embedding dimensions for different variety of tasks across a number of tasks (such as person re-identification). Therefore the authors have provided theoretical insights as to why such collapsing happens for learning a distance metric with hard negative sampling, and ways to overcome them by performing experiments across different datasets and different backbones.

**Strengths:**

The connection between network collapse with the batch size or embedding dimensions; using isometric approximation is novel and interesting.

The authors have provided a detailed analysis of their proposed approximation between the hard negative-mining triplet loss and the Hausdorff-like distance metric along with detailed proofs in the appendix.

The authors have provided a substantial number of experiments across different datasets and different network backbones to show the effectiveness of their proposed work.

**Weaknesses:**

The major weakness of the paper is only showing the effectiveness of their proposed method for hard negative mining strategies for learning the distance metric. There have a considerable number of new sampling strategies which improve upon hard negative mining, especially aimed to tackle the slow convergence rate and the network collapsing phenomena. A different variety of loss functions have also been developed  to tackle the above mentioned issues. Therefore the impact of this paper is limited by the choice of the single "hard mining strategy". The methods in [A, B, C, D] aim to reduce the dependency of the batch size or the size of the embedding. A similar study has also been conducted in [E]. These methods need not always provide any theoretical explanations (which is a big plus for this paper), but they attempt to solve and tackle similar issue (network collapse). So it will be interesting if such equivalence can be drawn with such methods, which is missing in the paper.

[A] W. Zheng, J. Lu and J. Zhou, "Deep Metric Learning With Adaptively Composite Dynamic Constraints," TPAMI 2023.

[B] Zheng, Wenzhao, et al. "Dynamic metric learning with cross-level concept distillation." ECCV 2022.

[C] Zhang, Borui, et al. "Attributable visual similarity learning." CVPR 2022.

[D] Zheng, Wenzhao, Jiwen Lu, and Jie Zhou. "Deep metric learning via adaptive learnable assessment." CVPR 2020.

[E] Musgrave, Kevin, Serge Belongie, and Ser-Nam Lim. "A metric learning reality check."  ECCV 2020.

**Questions:**

(1) Will such approximation hold true for any other sampling strategy or loss functions as mentioned in the Weakness section?

(2) In [A], the authors experiment with learnable $\beta$'s. Can the authors more some insights as to what would happen if $\alpha$ is also learned, instead of keeping it fixed.

(3) Have the authors done any experiments on face recognition datasets similar to [B, C]? It will be interesting to see if the proposed explanations hold true for experiments on face recognition too.

[A] Wu, Chao-Yuan, et al. "Sampling matters in deep embedding learning." ICCV 2017.

[B] Wang, Hao, et al. "Cosface: Large margin cosine loss for deep face recognition." CVPR 2018.

[C] Deng, Jiankang, et al. "Arcface: Additive angular margin loss for deep face recognition." CVPR 2019.

---

> ### Author Response · Authors · 2023-11-18
>
> We thank the reviewer for the constructive comments. We respond to the comments below and update the manuscript accordingly.
>
> **$\mathbf Q_1$: "The major weakness of the paper is only showing the effectiveness of their proposed method for hard negative mining strategies..."**
>
> $A_1$: Thank you for this insightful feedback.
>
> While our work focuses on hard negative mining on the triplet loss, it is worth noting that the theory of isometric approximation as used in this manuscript is independent of the triplet loss. Consequently, it can be applied to any system utilizing the Euclidean metric, for instance the pairwise contrastive loss ($\mathcal L = \|x-x^+\| - [\alpha - \|x-x^-\|]_+$) or the margin loss ($\mathcal L = [\|x-x^+\| + \alpha - \beta]\_+ + [-\|x-x^-\| + \alpha + \beta]\_+$). As both these loss functions essentially use a difference in distances, we can use the isometric approximation theorem to connect to other potentially useful functional forms which may reveal interesting theoretical conclusions.
>
> On the other hand, the fundamental reason for the choice of hard negative mining is that it resembles a $\max_{x,x^+,x^-} \mathcal L(x, x^+, x^-)$ type of operation, which via the isometric approximation theorem, can be directly connected with our defined Hausdorff-like metric $d_\textrm{haus}(f_1, f_2) = \max_x d(f_1(x), X_{f_2}^i)$ or $\max_x d(f_2(x), X_{f_1}^i)$ because they both use max operations.
>
> Following the unified metric learning formulation defined by Zheng et al. [A] in eq (7), a different mining strategy would require some change to the Hausdorff-like metric. In particular, we suspect that rather than $\max_x$ we would have some function dependent on the chosen graph $G(\mathbf Y, \mathbf P, \pi(\mathbf y, \mathbf p))$ and distribution $\pi(\mathbf Y,\mathbf P)$ from [A]. But it does feel possible that a connection is waiting to be made there.
>
> We thank the reviewer for this feedback. We have amended our conclusion with these insights and highlighted the changes in yellow.

---

> ### Author Response · Authors · 2023-11-18
>
> **$\mathbf Q_2$: "Will such approximation hold true for any other sampling strategy or loss functions as mentioned in the Weakness section?"**
>
> $A_2$: We thank the reviewer for the question. Continuing our response from the previous section, the theory
> of isometric approximation as used in this manuscript is independent of the triplet loss. Further, the sampling strategy and loss function are also somewhat independent in that one can freely swap the sampling strategy or the loss function. For instance, if the isometric approximation holds for a given combination of sampling strategy A + loss function B, we expect it to hold for hard negative mining + loss function B and also for sampling strategy A + triplet loss.
>
> **$\mathbf Q_3$: "Can the authors more some insights as to what would happen if $\alpha$ is also learned?"**
>
> $A_3$: The $\beta$ in [A] fulfills a different role from the margin parameter $\alpha$ in the triplet loss for our paper. While we have not performed experiments with learnable $\alpha$, we expect that because $\mathcal L_{\alpha=0}(x) \leq \mathcal L_{\alpha}(x)$ for all $\alpha > 0$, the network would always learn $\alpha=0$. Increasing $\alpha$ can never decrease the loss value, so naturally, it should always go to zero without some forcing term. It is worth noting that the margin $\alpha$ in [A] exhibits a similar behavior and is also not treated as a learnable parameter. Since we do not have a boundary term $\beta$ in our formulation, we hope that our response on learning the margin parameter $\alpha$ addresses the query above.
>
> **$\mathbf Q_4$: "Have the authors done any experiments on face recognition datasets..."**
>
> $A_4$: We thank the reviewer for the suggestion. Our recent investigation with the face recognition dataset from [B, C] (Labeled faces in the Wild) reveals a trend similar to the proposed explanations, where the likelihood of network collapse increases with an increase in the batch size. Specifically, the final embedding diameter of 2.10 with a batch size of 4 decreases to a value of 0.68 when the batch size is increased to 512, indicating network collapse.

---

### Author Response · Authors · 2023-11-17

We appreciate the insightful and constructive feedback provided by the reviewers. The comments have been helpful in refining and improving the overall quality of our manuscript.

The changes to our revised manuscript are highlighted in yellow and are summarized below.

- **Introduction:** We have added a reference to a related work as suggested.
- **Figure 2:** We have amended the figure caption to more accurately describe the figure. The examples in the following section should be easier to understand now.
- **Figure 3:** We have simplified figure 3, aligning with the reviewer's suggestions. The illustration of difference in distance metric is clearer now.
- **Figure 4:** The figure has been re-designed to show the sequential collapse of a set of points, annotated with distance values for clarity.
- **Discussion:** Drawing from the insightful comments from [QYwJ], we have added a paragraph to the Discussion section on future work and our theory's applicability to other metric learning losses and triplet mining strategies.
- **Appendix / ResNet-50 Experiment:** We have added an additional experiment using a ResNet-50 backbone, as per reviewer's suggestions.

---

### Meta-Review · Area_Chair_bRtp · 2023-12-10

**Metareview:**

This paper establishes a connection between network collapse and hard-negative mining in metric learning using the theory of isometric approximation. Based on the developed theory, the authors show that the use of large batch sizes in hard-nagative mining could lead to network collapse. The submission was reviewed by four experts in metric learning and received a mixed score. The main criticism is the lack of comparisons and insights about more advanced mining strategies. The authors, while not explicitly looking into other strategies, stated that the developed theory could be used there as well. Since hard-negative mining is a very well-established paradigm (even beyond metric learning), the AC recognizes the significance of the developed theory and, therefore, recommends accepting the paper.

**Justification For Why Not Higher Score:**

The paper does not suggest beyond hard-negative mining, and as such, I do not think a spotlight/oral presentation is applicable here.

**Justification For Why Not Lower Score:**

I am not familiar with a similar study and the developed theory seems novel to me and very useful.

---

### Decision · Program_Chairs · 2024-01-16

Accept (poster)